# Team-Based Learning Experiences of Nursing Students in a Health Assessment Subject: A Qualitative Study

**DOI:** 10.3390/healthcare10050817

**Published:** 2022-04-28

**Authors:** Hyung-Ran Park, Eunyoung Park

**Affiliations:** 1Department of Nursing Science, College of Medicine, Chungbuk National University, Cheongju 28644, Korea; hyungran@chungbuk.ac.kr; 2College of Nursing, Chungnam National University, Daejeon 35015, Korea

**Keywords:** nursing students, qualitative research, team-based learning

## Abstract

Health assessment as a subject comprises knowledge and practices in which health problems are identified by collecting individual health data. As the subject requires fast learning of voluminous content, it becomes cumbersome. Team-based learning (TBL) has been proposed as an effective teaching and learning strategy in such situations. This study aimed to explore the lived TBL experiences of nursing students from their perspectives in a health assessment subject. This study adopted a qualitative research method. TBL was applied in a health assessment subject at a university in South Korea, as a 2-credit course for 16 weeks. Twelve sophomore nursing students who enrolled in a health assessment subject were the participants. Data were collected through individual in-depth interviews and analyzed using Colaizzi’s method. The results contained twelve themes categorized into five theme clusters: “Getting ready for learning”, “Effective class promoting concentration and immersion”, “Proactive participation in a non-hierarchical learning environment”, “Complementary collaboration”, and “Sense of burden”. The findings suggest that TBL is an effective teaching and learning strategy for the discipline, imparting positive experiences such as class engagement, teamwork, learning outcomes, and improvement of problem-solving skills if students’ role conflicts and continuous learning burden can be addressed.

## 1. Introduction

Knowledge and practice education are essential parts of the curriculum of nursing courses. Instructors are expected to facilitate the development of a professional thinking process among students for learning and exploration, provide opportunities to apply the knowledge they have learned, and equip them with the skills and attitudes required to cope with various clinical situations [1].

Health assessment, comprising knowledge and practice in the undergraduate nursing course, is a compulsory subject that collects health-related data from patients of nursing care and identifies potential health problems. Students’ adequate understanding of prerequisite subjects such as anatomy, physiology, and pathology is required for its application in a health assessment subject, as they learn about normal and abnormal health status. As a health assessment subject in the Korean nursing curriculum combines knowledge and practice and is usually run as a 2-credit course, the subject contains enormous content to be learned within a few hours allocated, causing further difficulties for students [2,3].

Team-based learning (TBL) has been proposed as an effective teaching and learning method to address the practical difficulties of learning a large amount of content within limited hours allocated [4]. TBL progresses through several stages. In a pre-class readiness preparation stage, students study using learning material such as textbooks, reading materials, or videos; during a readiness assurance stage, individual readiness assurance test (iRAT) and team readiness assurance test (tRAT) are performed using multiple-choice questions focusing on the core concept, and a consensus-building discussion in tRAT improves interaction and decision-making process. Through a feedback and mini-lecture stage, an instructor reviews the core concept tested in iRAT and tRAT. Then, students are prepared to solve a complex problem in the next application exercise. Subsequently, in the application exercise stage, a complex problem is solved by the team, and all teams are provided the same problem that they can face in real-world workplace [4]. The reported benefits of TBL include enhanced learning effect and development in capabilities of interactions and interpersonal skills [4]. The students’ capabilities such as communication, collaboration, and problem-solving, and learning attitudes such as satisfaction and engagement are positively influenced by TBL [5]. However, there have been few studies on the essence of the TBL experiences from the students’ perspectives, which limits the understanding of the TBL experience in the actual subjects of learning [6,7,8,9,10,11,12].

The existing qualitative research on TBL experiences from students’ perspectives has been conducted on students of multidisciplinary courses [6,7,8,9] and not exclusively on a health assessment subject. Some studies have only performed an analysis of short responses obtained as a part of research conducted along with quantitative research [10,11] and thereby have only limited information on TBL experiences. A previous study [12] on TBL experiences of international nursing students presented difficulties they experienced due to the cultural differences but not students’ experiences of TBL alone. Although the benefits of TBL have been reported in terms of learning outcomes and indicators of clinical capacities performance, it is important to investigate the phenomenological essence of TBL experiences from students’ perspectives to determine the effects of TBL as a teaching and learning strategy.

A qualitative study on students’ experiences of TBL should be performed in their natural context and environment as an overall understanding and insight on the research topic can be acquired [13]. Phenomenological methods ensure an understanding of how participants of the study perceive phenomena and how such perceptions influence their behavior [14].

Therefore, the purpose of this study was to conduct an in-depth exploration of the lived TBL experiences of nursing students in a health assessment subject. The research question was “How do nursing students perceive their TBL experiences in a health assessment subject?”.

## 2. Materials and Methods

### 2.1. Design

This qualitative study was based on a descriptive, phenomenological approach for understanding the learning experiences of nursing students in a health assessment subject with the application of TBL. In the descriptive phenomenology influenced by Husserl’s philosophy, it is important to explore essential structures of the phenomena [15]. This study complies with the comprehensive checklist of the Consolidated Criteria for Reporting Qualitative Research (COREQ) [16].

### 2.2. Participants and Setting

Eligible participants were sophomores who enrolled in a health assessment subject at a nursing college in South Korea and who were willing to join this study. To minimize the researcher’s effect, we excluded students who were taking classes of another subject taught by the researcher at the time of the recruitment. Purposive sampling was used to select those who could best represent phenomena of the relevant topic [17]. Twelve students out of a batch of 72 were finally selected as study participants; they were recruited based on the composition ratio by sex, age, self-perceived personality, and grades. Health assessment, which applies TBL, is a two-credit subject, with 50-min of lecture class and 100-min of laboratory practice per week, for 16 weeks in total. All second-year nursing students were required to take a health assessment subject involving TBL at the college. There were twelve teams, each comprising six members. The lecture topics included the respiratory, cardiovascular, gastrointestinal, and nervous systems. After forming teams, the TBL classes were conducted, keeping an order based on pre-class preparation, iRAT and tRAT, feedback and mini-lecture, and application exercises according to Michaelson’s [4] guidelines (Table 1).

### 2.3. Data Collection

To minimize factors that could negatively impact the voluntary nature of participation, the study was completed after the semester was over and grade assessments were finalized. The face-to face, semi-structured, and in-depth individual interviews were conducted by the corresponding author, who is an experienced qualitative researcher and was not involved in the health assessment subject and was not responsible for evaluating student performance and grades. All interviews were conducted in her office at the school, which ensured a quiet environment. Twelve participants were sampled until reaching saturation [17]. The duration of the individual interviews lasted from 57 to 84 min. One interview was conducted for each participant, and additional interviews were conducted for two of them. All interviews were recorded in MP3 format, and the author noted any remarkable information in terms of atmosphere, content, or expressions. The recorded interviews were transcribed by the research assistant, and the transcripts were verified by the participants. The main interview question started as follows: “While you were taking the health assessment course, what kind of experiences did TBL offer you?” Further, specific questions were asked based on the students’ answers and responses. Table 2 provides examples of the key interview questions.

### 2.4. Ethical Considerations

This study was conducted upon the approval of the Institutional Review Board of the university (AJIRB-SBR-MDB-13-350). Considering that the participants are students who opted for a course offered by one of the authors, the processes of obtaining their consent and conducting interviews were carried out by the other author who was not in charge of the course. Participants were briefed about the purpose and procedure of the study, the right to withdraw from participation in the study, and the policy on confidentiality and anonymity. Subsequently, they signed the informed consent form declaring their volition for participation in the study. Participation in the study had no impact on the grades of participants since they were already evaluated. To protect vulnerable participants, we excluded students who were taking classes of another subject taught by the researcher at the time of the interviews, which could be an obstacle to voluntary participation. Participants were given a gift certificate as a token of appreciation for their participation in the study.

### 2.5. Data Analysis

Data were analyzed based on Colaizzi’s phenomenological method [15] using NVivo 12 Plus for Windows software (QSR International, Burlington, MA, USA). In the first step of the analysis, the collected data were reviewed. To obtain the proper impression of the participant’s statement without compromising the meaning expressed, the transcript was read repeatedly. Subsequently, we reviewed each transcript to extract significant statements pertaining to the experiences of TBL in a health assessment subject. In the third step, redundant expressions were excluded, and meanings were formulated in abstract forms after carefully examining derived statements. Efforts were made to understand the intended meanings of the participants and construct meanings in general forms and wordings. In the fourth step, the meanings were categorized into themes and theme clusters. Significant statements and generalized statements were grouped to organize the themes. Further, we underwent the process of continuous reflection on the original data and reconstituting the themes and theme clusters to examine whether they concurred in meaning making. In the fifth step, themes and theme clusters were comprehensively explained and described. Moreover, in the sixth step, to construct the essential structure of the experience, explanations of themes and theme clusters and the original data of the participants were presented in detail. The essential structure of the TBL experiences of the participants in the course was described. Finally, the themes, theme clusters, and quotations of the results were shown to two interviewed participants, and the consistency of the results with their own experiences were validated.

To ensure the rigor of the results derived in this study, the process was followed [18]. We established credibility by receiving confirmation from the participants on the themes and comprehensive descriptions. In-depth data were elicited, and the phenomena were described correctly, ensuring transferability. The interview schedule and situations were recorded using field notes and memos, and all data were recorded and transcribed to maintain dependability. To achieve confirmability, the corresponding author who conducted the interview tried to practice reflexivity for eliminating bias during the interview. Each researcher (the first and corresponding author) also independently analyzed the collected data to flag inconsistencies in the wording of the themes and construction before consensus on the final result.

## 3. Results

There were eleven female and one male participants, aged 20–23 years (median: 21.5 years), and all participants had no previous experience of TBL (Table 3).

The results can be presented through five theme clusters and twelve themes, and the theme clusters are as follows: getting ready for learning, effective class promoting concentration and immersion, proactive participation in a non-hierarchical learning environment, complementary collaboration, and sense of burden (Table 4).

### 3.1. Getting Ready for Learning

The pre-class preparation and readiness assurance tests, which are essential to TBL, motivated the participants to learn properly. Pre-class preparation also created a positive atmosphere of learning for the entire class.

#### 3.1.1. Constructive Sense of Burden That Encourages Learning

The students felt that the assignments of pre-class preparation or the readiness assurance test created an adequate level of internal tension for learning and a “constructive sense of burden”, which motivated their studies. As preparations were compulsory for TBL, they became consistent in studies, which was change in their habit that they appreciated.

*Constructive sense of burden? As the sense of burden had a positive influence, I became industrious, and although the word burden may sound negative, it was not in an unacceptable level. (omitted) I was positively prompted to continuously take the test during the course so as not to let go of the tension. This habit has now set a positive pattern in my studying*.(Participant 4)


*I hardly study during the non-exam period, but this was a class that encouraged me to study consistently.*
(Participant 10)

The subject includes voluminous learning content and practicum. However, through the pre-class preparation in TBL, students could understand and take in what they have learned in the class. That is, as TBL was effective in terms of their studying, students commented that it provided them a chance to appreciate the importance of pre-class preparation. Subsequently, they have developed a sense of reward and achievements for independently studying hard and also were motivated to apply this method for studying other subjects.


*As we had already studied the content once during pre-class preparation, it was nice to be able to answer the professor’s questions when asked. I used to take classes without knowing anything about the content. So, it caused a positive change in my behavior as I followed the pattern of preparing for the class in advance. I felt a sense of self-satisfaction and accomplishment with my effort, irrespective of my Grade. With the preparation, it became easier to understand the content during the class, which also contributed to my sense of achievement. This was the first subject that gave me such an experience and feeling.*
(Participant 9)

#### 3.1.2. Readiness in Learning Attitude and Environment

Participants acknowledged that TBL created their readiness in learning attitude and environment. Unlike in other classes, where the atmosphere was noisy and distracting before the session commences, with students arriving just in time for the class, participants said that for the health assessment classes, most students arrived well in advance, forming a positive atmosphere. Furthermore, participants felt that the class set a tone of encouragement toward fellow teammates to be assiduous rather than competing against one another.

*We made a promise to the team members to arrive in advance, take a quiz with each other, and prepare for the class. I think the class was helpful because we were certainly more prepared for the tests, and the habit of reviewing right before the test really mattered*.(Participant 4)


*I think that those who are always late for class are disruptive to its flow. However, in this class, when we first set rules for our team, we decided not to take out our cell phones during class, arrive five minutes earlier (before the class), etc. For other classes, nobody expected students to arrive earlier, but for this class, students arrived even ten minutes earlier. Further, the atmosphere in the class was not like “Oh, she is studying”, but “Let’s do it together” (and it was good).*
(Participant 11)

### 3.2. Effective Class Promoting Concentration and Immersion

In TBL, as the nursing students repeated the systematic cycle of pre-class preparation, quizzes, lectures, and application exercises, they found the contents easier to comprehend. The method also enabled concentration and immersion into the class through continuous pre-class preparation and review, compared to the traditional lecture-driven classes. The students felt that the effectiveness of TBL improved their learning ability and problem-solving skills.

#### 3.2.1. Identification of Key Learning Points

Owing to the pre-class preparation, the students had an understanding of key learning points during the class; it prompted them to listen more intently. In particular, the students appreciated how TBL made them focus on the key contents during the class, which became helpful during the time of practicum.

*As the lecture was focused on key points, it was easier for us to follow the class. We could comprehend the lecture irrespective of its pace. With TBL, I could understand and remember which parts were important due to the quizzes, and this was also helpful for the practice of assessment*.(Participant 3)

*Preparing for the next class helped me grasp the key points of the class, and focusing on them was also helpful for the preparation for exams later. For example, I had the prior knowledge that “These points are important in the gastrointestinal system”, which allowed me to prepare better for exams, and this was also helpful in terms of performing health assessments for the practicum part. I could remember the essential points for the assessment*.(Participant 6)

#### 3.2.2. Repeating Patterns for Systematic and Continuous Learning

Participants highly valued the repeating patterns in TBL through pre-class preparation assignments, discussions with team members for the readiness assurance test and the feedback from the professor, and the key points in the mini-lecture. Unlike the conventional style of one-way communication of knowledge, through the systematic cycle of TBL in which the preparation and review are conducted in each stage focusing on the key points, the students could identify the parts that needing improvements and make extra efforts accordingly. Furthermore, TBL was also helpful in making students remember the content longer.


*For each class, the pre-class preparation assignments worked well. As I became familiar with the content through key points in the pre-class preparation, discussions for the team quiz session, and the lecture by the professor, I found it much easier to follow and study the content of the class. The best part of the class was that I studied the content repeatedly. In other subjects, before I study for the exam, the lectures I attend for the main class are the only input I have. As the subject of Health Assessment requires both knowledge and real-life application, I need to continue to learn the content until it is properly understood.*
(Participant 8)


*Unlike the rote-learning-type classes, I felt that I could review the whole class. (For other classes), I felt that I only had inputs, but in this class, I liked the fact that the class was structured in a way that ensured input, output, and feedback, which facilitated a smooth connection between each element. Because of the structure of TBL, I have to do pre-class preparation assignments once a week, so this reminded me of the key content. So, the cycle of pre-class preparation, taking a quiz, and the main class was really helpful. I became familiar with the style of the class and felt that I had learned more. It was also easier to review and summarize what I have learned.*
(Participant 11)

The students found the TBL method beneficial to study consistently with better concentration, which helped them improve learning abilities and satisfaction.


*As I have been studying the content systematically, I received a better grade considering the time I have spent on this subject, compared to other subjects, and I could understand the content more efficiently. For me, the more I understand, the more I find the subject interesting. The pattern of this class allowed me to build my understanding of each chapter rather than doing it all at once. This is something that I scarcely do on my own. I usually tend to postpone the review or study, but this type of teaching method encourages me to study consistently.*
(Participant 9)

#### 3.2.3. Improvement in Problem-Solving Abilities through Reasoning of Evidence

Students appreciated the approach of reasoning evidence for problem-solving through team discussion during tRATs and application exercises, as it improved their practical problem-solving skills, which took place at real-life workplaces, beyond that during their individual studies. They also mentioned that preparing the evidence for problem-solving prompted them to study harder throughout the semester and not just during specific times. In particular, they mentioned that application exercises were important in realistic and practical learning for application of health assessment skills as nurses, through critical thinking during team discussion.


*I had the opportunity to find the relevant evidence, think it through, and select the answer based on a valid reason. Before this class, mostly, I chose the answers through rote learning rather than based on clear evidence and reasons.*
(Participant 8)


*When I study individually, I mostly tend to accept the information without critical analysis. However, in this class, I think to myself first before I express my opinion to the team members, and after listening to their different thoughts, I reconsider my thoughts and reasons. When they seem to have better reasons, I think about my reasons again. In this process, I think I have improved my critical thinking skills.*
(Participant 4)

As can be seen from these examples, the students pointed out that TBL was a teaching method more suited for the subject, as it helped them learn the content thoroughly and apply it in their practicum for the problem-solving process.


*Unlike the times when I learned content in theory from books, when I actually had the practical sessions, I learned to be flexible and adaptive. As I went through the course, I realized that in the subject of Health Assessment, practice is as important as theoretical knowledge, and I was involved in interactions with other people or exchanging opinions for problem-solving. Practical capabilities are really important for this subject, and the method adopted nurtured them.*
(Participant 4)

### 3.3. Proactive Participation in a Non-Hierarchical Learning Environment

Compared to the conventional lecture-driven classes, where students were mainly passive listeners, TBL allowed the experience of various exchanges of opinions and feedback, creating a non-hierarchical environment.

#### 3.3.1. Differentiation with Novelty and Fun

Although students who were familiar with the traditional type of lecture-centered classes were initially intimidated with the innovative method of TBL, they evolved to find it interesting, which involved active learning through the process of pre-class participation, team quizzes, and discussion/debate sessions.


*I initially felt pressure as I knew little about TBL. I had been used to the rote-learning-type classes where the professors teach and we listen. However, with TBL, I did not study on my own but was given many opportunities for discussion and sharing thoughts during the class, which was more fun. I strongly felt that this class was different and special compared to other classes.*
(Participant 9)

As can be seen from the testimonials, some participants identified TBL as the method they had envisioned as the ideal image of a university class.


*TBL... I think this is the type of college class I have envisioned before. Well, we have plenty of chances to exchange our thoughts and opinions, feel close to the professors, and actively exchange ideas with them.*
(Participant 4)

#### 3.3.2. Flexible and Comfortable Class Atmosphere

The participants felt the learning environment with discussion in the TBL class liberating compared to the lecture-type classes. In particular, they commented that TBL class formed a non-hierarchical and harmonious environment. As the rules were set among team members, there was mutual respect in the discussion, and students properly listened to one another. Students willingly played their parts and tried their best without any free riders.


*In the curriculum for the department of nursing, apart from practicum sessions, it felt like a continuation of high school days (an expression indicating traditional rote-learning based education). However, with TBL, I felt some kind of freedom…I think the team I belonged to have equally shared the roles and responsibilities. We all participated equally without burdening anyone particularly. At the time of team discussion, as all of us could make claims or give opinions without restrictions, we felt equal.*
(Participant 6)


*All of us voiced our opinions. There was never a time when any of us refused to answer or did not express their opinions when we were asked to do so. I thought that there was no passive student in particular; we voluntarily expressed our thoughts. All of us paid attention while listening to others’ opinions, trying to understand their reasoning…*
(Participant 9)

#### 3.3.3. Active and Diverse Interactions

As several activities needed to be conducted with other people, students were encouraged to actively express their thoughts and exchange different ideas, which also served as an opportunity to understand others’ perspectives. There were active interactions during the time of feedback from the professor and team discussion. The students mentioned that this was different from other group assignments or presentations where they simply divided their roles and hardly had exchanges between team members. Participants felt that the communication competency of the introverts and extroverts in their batch had improved.


*TBL had many team activities. As there were many people, we collected different opinions, which facilitated active communication. We talked together, and if anybody did not know anything, we helped them with the information we had. I think we had an appropriate amount of talks and discussions.*
(Participant 1)


*As for group assignments in other classes, there were students who did not did not actively participate but only performed delegated roles. However, with TBL, the team discussion time when we all voiced our opinions was really different from other occasions. As I had not had opportunities for active discussions or debates, I liked this experience.*
(Participant 5)

Active interactions took place not only among students but also with the professor, and students found getting immediate feedback beneficial.


*Usually, after taking quizzes, we do not get to know our results on the spot. (With TBL), it was good that we could get our results and discuss and reason our answers with the professor and other teammates. Before TBL, one-way communication in classrooms often caused the students to miss the message, but there was no such case in this class.*
(Participant 2)

### 3.4. Complementary Collaboration

Participants thought that the accountability for the team as a team member did not pressurize them but motivated their individual studies too. With recognition of the accountability for the team, participants worked in cooperation to submit their team assignment through teamwork. Through the collaboration process, the team members developed intimacy and a sense of belonging and unity as a team.

#### 3.4.1. Accountability for the Team

The participants felt obliged to the team while studying contents for the class. Hence, they did not procrastinate or find excuses. They felt accountable to the team, which motivated them to work harder and contribute as other hard-working team members.


*If I don’t study, I become a nuisance to the team. So when I do pre-class preparation, I make sure to learn well as my teammates will hate it if I hardly know anything! That thought prompted me to study harder. Strangely, I do not find it burdensome but motivational to my studies.*
(Participant 7)


*Perhaps I could put it as a developing sense of accountability…? Accountability within the team manifests as my respect toward team members who worked really hard. So regardless of the grade, I think it would be undesirable if I hardly make proper contribution to the team, so I try to work harder…In this way, accountability was developed, but I had less individual burden, and it helped me study better.*
(Participant 8)

#### 3.4.2. Teamwork: A Sense of Being “One Team”

Participants preferred the method as it gave them opportunities to help each other out and complement other members’ weaknesses through teamwork in team quizzes and application exercises. The students pointed out that when the team worked together to submit a team assignment with quality, they could learn from one another. As they shared information and developed an emotional rapport, they overcame their prejudices about fellow members and built intimacy between them, which gradually developed into team spirit.


*While we were taking quizzes or conducting physical examinations within the team, we learned from each other. It would have been highly challenging to do this all on my own, but when this was conducted by the entire team, we could share information, build rapport, and develop intimacy.*
(Participant 1)


*The five of us had to come together and eventually become one team. So in the process, I often felt that we all contributed in producing good work. When we all work together to produce an output, it is the product of our combined effort. I liked this aspect of the class. During practicum sessions, we got together more, helped each other out, and exchanged information among team members.*
(Participant 2)

### 3.5. Sense of Burden

Although students had beneficial experiences in the TBL method, the compulsion to contribute as a team member may be burdensome to those who are not expressive. Pre-class preparation and various class activities required from TBL became a learning burden for some students.

#### 3.5.1. Burden from the Role

The participants explained that the TBL class might be difficult for those who are introverts or reserved to give opinions during discussion or debates. Often, a team member with active personality took the initiative to lead the team. It would burden that person and may impact other passive and introverted members negatively.


*There are some introverts who don’t present opinions in discussions. They continue to remain as they are. Their contribution remains trivial in comparison with that of the active members.*
(Participant 3)

One participant felt this role burden more, especially at the beginning of the class, when team members were not familiar with the discussion.


*Our team needs to have a good atmosphere to practice well, so I was worried that I would make the atmosphere in our team turn ugly if I didn’t talk by myself or talked too much. So I had to coordinate all of team members’ opinions and pay attention to them. It wasn’t hard, but it was uncomfortable at first.*
(Participant 12)

#### 3.5.2. Learning Burden

The students expressed that it would be burdensome if the method were to be applied to multiple subjects as pre-class preparations and class activities of TBL demand a considerable amount of work. They were also skeptical of its effect in the early stages of the class in terms of them internalizing habits such as self-directed learning and incorporating it into their routines as the course does not bind them to do so. Some students preferred conventional lectures due to the learning burden from pre-class preparation of TBL.


*I could say that initially, I used to get annoyed about pre-class preparation… Before I realized the effect of the preparation, I had preferred lectures instead of doing the assignments. Preparing for the quiz is a difficult part, too. It certainly has positive sides, but I always felt the pressure on the day before Thursday (day of class) due to the scheduled quiz.*
(Participant 11)


*Generally, it was okay, but because of the pressure to take the quiz…well, just like all quizzes are burdensome, this one was no exception. I tried to be persistent because I know that TBL is useful. However, I prefer lecture-type classes as they are less burdensome…*
(Participant 5)

## 4. Discussion

This study was conducted to understand the TBL experiences of nursing students who had enrolled in a health assessment subject. The students expressed that the TBL experience prepared them for learning and that it was an effective class that promoted concentration and immersion, encouraged them to proactively participate in a non-hierarchical learning environment, and induced complementary collaboration. However, in the process of discussion or debates, they felt burdened with the responsibilities of their assigned role.

The first theme cluster was “getting ready for learning”. The pre-class preparation makes the students recognize the importance of learning on their own and develops positive motivation toward voluntary participation in learning [19]. In the teaching strategies of TBL, the preparation material provided before class is designed to help the students in their in-class learning and discussion [20]. This is due to the fact that the process of having prior knowledge and activation of the knowledge is essential to learn new content [19]. Various types of tests for which students prepare based on accountability induce them to develop learning attitudes [11]. Students spend more time in learning in TBL than in the normal lecture. In the process of taking the tests, they become reflective of their existing learning attitudes, thereby developing positive autonomy for learning [19,21]. The consistent tests make it impossible for the students to remain passive in the learning environment [19], and classmates participating in TBL introspect and form a learning environment with positive attitudes [21].

Health assessment is a subject based on the knowledge of human bioscience [22]. Therefore, applying a pedagogical strategy to acquire knowledge prior to practice is expected to enhance learning readiness. In the process of pre-class preparation, the instructor should consider a well-designed test process to facilitate the development of a learning attitude by ensuring experiences of positive and proactive learning.

The second theme cluster was “effective class promoting concentration and immersion”. An outcome-based approach identifies the TBL effect in terms of increased level of concentration and effectiveness of the class experienced by students [5]. In TBL, reading material and RATs are designed to reinforce knowledge focusing on the core concepts. Application activities are designed to improve competencies in critical analysis and problem-solving skills by applying core concepts to significant problems in real-life workplaces [4]. In previous studies, students who participated in TBL found the method effective in terms of understanding core concepts, which helped them concentrate better in class [23] and identify the gap in their knowledge [11]. In application activities, at times of presenting opinions within a team and among teams, students were required to explain their decisions and advocate their positions with accountability [21,24]. It enhanced their capabilities of decision-making or finding evidence to support their reasoning. This is due to the fact that the class immersion of the students is increased by realistic case-based problems, and critical thinking skills are developed in the process of concentrating and discussing with team members [23]. TBL is described as a constructive process that includes several stages [5], whereby students experience systematic learning with repeated exposure to core concepts. Previous studies have shown that TBL composed of these stages serves as an effective experience in helping students learn consistently by preparing for assignments and tests [10].

In a health assessment subject, to correctly recognize and classify the collected physical examination data into normal and abnormal findings, it is important to possess knowledge in human bioscience [22]. Therefore, TBL, with its method of repeatedly learning core concepts, is considered to induce the concentration of the students to increase the efficiency in learning. As nursing education focuses on the interpretation of significantly abnormal clinical findings rather than normal examination findings [2], applying TBL to identify, analyze, and solve the significant problem based on clinical reasoning will serve as an effective pedagogical strategy to apply practical health assessment skills and interpret the findings.

The third theme cluster was “proactive participation in a non-hierarchical learning environment”. A learning environment in a non-hierarchical atmosphere where mutual opinions are respected with active interactions encourages students to participate in activities [19]. When students first experience a new teaching method, they become interested in its novelty when they recognize that there is a difference in the method of knowledge transfer compared to the traditional lecture-oriented class. This experience was also reported in previous studies [9]. In particular, in subjects to which traditional lecture-style teaching methods have long been applied, such as physiology or anatomy, students found TBL to be an enjoyable experience [8,9]. In TBL activities, in the process of exchanging opinions, students actively listen to one another with mutual respect, creating a comfortable and harmonious learning environment [6]. Such a team environment allows students to express their opinions freely and participate actively in discussions, enriching their learning experience [11]. Previous studies have reported that active interaction is one of the best opportunities in the TBL experience [10,11,19]. The interaction between students and the instructor through feedback creates a non-hierarchical learning atmosphere compared to traditional lecture-based methods [23] and encourages students to participate actively in the learning [19].

In nursing education, exploring pedagogical methods suitable for preparing a health assessment competency is a constant challenge [22]. Thus, it is considered that the teaching method of TBL in which students are encouraged to learn with interest will be effective. As health assessments take place based on interactions with patients, well-trained interactions through TBL can help prepare students for future nursing practice.

The fourth theme cluster was “complementary collaboration”. In TBL, complementary collaboration can be described as the process in which the students play cooperative roles to promote other members’ learning by recognizing their learning needs [6]. The building blocks of TBL motivate students to engage in learning and encourage teamwork. As students think about their contributions to tests and teamwork, they develop a sense of accountability, which often leads to consistent learning [25]. In previous studies, it was also shown that students experienced a sense of accountability due to the preparation required for team activities and spent more time in learning [11]. In particular, teamwork as a learning tool enhances students’ active participation and interactions in discussions, helping them to develop mutually respectful relationships [25]. This type of team collaboration allows the team to trust each other and develop team dynamics [8] and prepares team members’ capabilities and their products for success [6]. In previous studies, students who participated in interprofessional TBL with courses such as nursing, medicine, and physical therapy found the teamwork experienced in TBL instrumental for collaborative work in their future vocational activities [6,8,9].

Health assessment is a core subject in the curriculum that serves as a basis for evidence-informed care in studies related to nursing and healthcare [2,19]. Therefore, it is meaningful for the students in the healthcare field to experience TBL to learn and practice roles in collaboration and develop team dynamics.

The final theme cluster was “sense of burden”. In the process of TBL, while the students experienced the effect in terms of TBL learning outcomes, they also experienced the burden of their role as team members and the extent of learning. When students are not prepared for activities in TBL or have team members who are not interested in those activities, there will be differences in their level of knowledge [11]. The unprepared or uninterested members will not be able to contribute to team performance, which will disturb the team dynamics [10]. Some students were not being able to present their opinions freely. A formal leader is assigned to communicate the team members’ decisions, and an unofficial leader who actively presents opinions with high motivation and aspiration for successful completion of team activities may emerge in the process [7]. Further, in some cases, the team dynamics is dependent on the opinions of students who are trusted in terms of their knowledge; in this case, there will be students who remain silent or passive even if they have different opinions [9]. As mentioned in previous studies, students feel burdened in the process of participating in team discussions in TBL, which necessitates the consideration of various characteristics of team members before forming a team or appropriate intervention by the instructor in team discussions [10].

Other types of burden have been reported in terms of the time spent for pre-class preparations, tests, and group work and interactions during application exercises. Students are often perplexed due to the scope and depth of learning materials [11] and think that the time spent for all of the preparation and tests may exceed that of the traditional lecture-oriented method [10]. Students who have received traditional education and are not prepared for this type of student-centered, self-directed learning may be at the risk of failing to achieve learning outcomes in their individual learning process [10]. Therefore, it is necessary for instructors preparing for TBL to carefully consider the principles of each stage, referring to the experiences of burden and pressure reported by students to induce the desired learning outcome [24].

Health assessment is a subject that combines knowledge and practice. Thus, considering the burden in terms of team roles and the extent of learning, the instructor needs to ensure that teams are formed after reflecting on the characteristics of team members. Moreover, to solve the learning burden, the instructors must consider the time spent by the student, learning materials’ depth, and student’s learning history before constructing each phase of TBL. Thus, it will be possible to reduce the burden and pressure experienced by students and effectively ensure their participation in TBL.

### Limitations

In this qualitative research, experiences of twelve Korean nursing students were explored and analyzed in detail. However, there may be limitations in generalizing this study if applied to different cultures and settings. Nevertheless, the findings of this study are expected to provide effective guidance for instructors who plan to adopt TBL in a subject comprising knowledge and practice. A professor in charge of the subject participated as a researcher in this study, but her involvement was strictly limited to the process of analyzing the results. The process of acquiring informed consent and other processes such as interviews that require face-to-face meetings were conducted by another researcher to ensure unrestricted ions of the participants. Finally, this study has a limitation that we could not exclude the contamination of participant’s experience by the professor’s evaluation of this health assessment subject since participants were recruited after the end of the course evaluation.

## 5. Conclusions

This study is qualitative research with a phenomenological approach that investigates the TBL experiences of nursing students who have opted for a health assessment subject. The findings of this study may guide the instructors in creating the class environment for learning, planning the number of assignments, scheduling, team formation, and considering students’ perspectives at each stage of learning. If the burden on the roles and amount of learning content are reduced, the application of the TBL method is expected to enhance learning outcomes as well as learning competencies such as problem-solving, collaboration, and communication. It would also improve learning attitudes such as engagement in class and learning satisfaction.

## Figures and Tables

**Table 1 healthcare-10-00817-t001:** Stage-specific procedures of team-based learning applied to the respiratory system.

Stages	Specific Procedures
Pre-class preparation	In the material for the pre-class preparation, the key points of the lesson were provided on the front and back pages of the A4 paper based on the desired learning outcomes of each class, including normal and abnormal findings of anatomy, physiology, and health assessment. Pre-class preparation for each session took approximately 30–45 min, and written materials on the topic were provided a week in advance for all students. Example: Assessment knowledge—Inspection, palpation, percussion, and auscultation of thorax and lungs
Readiness assurance	Individual RAT (iRAT) and team RAT (tRAT) were conducted for each class; each test contained five multiple-choice questions with a scheduled duration of five minutes. Example: Questions about the shape of chest in inspection, tactile fremitus in palpation, level of diaphragm in percussion, vesicular sound in auscultation, and rhonchi as an abnormal finding were used.
Feedback and mini-lecture	Immediately after the readiness assessment, feedback was provided through team discussions and student-professor interactions following a lecture for 20–30 min by the professor, focusing on the core contents of the target topic. Example: Mini-lecture was provided using power point slides and health assessment video about thorax and lungs.
Application exercises	First, students collected information on the location, size, extent, and responses of the organ system with normal health assessment findings obtained from healthy team members. Subsequently, a health assessment was conducted on the abnormal module using a high-fidelity simulator, and a complex problem-solving skill was performed to achieve the defined learning outcome. Example: Normal findings—Assessment about respiratory history, inspection, palpation, percussion, and auscultation of posterior chest of team members was performed. Abnormal findings—One question with 5 choices about adventitious lung sounds using a high-fidelity simulator was provided to teams and they presented the process and rationale of the answer.

RAT, readiness assurance test.

**Table 2 healthcare-10-00817-t002:** Examples of key interview questions.

While you were taking the health assessment course, what kind of experiences did TBL offer you? How did you feel about the process of preparing for TBL? What is a useful experience you have availed with TBL? What are the difficult or disappointing experiences you have had during TBL? What were the differences between the health assessment class following TBL and other classes? What were the changes you have experienced before and after TBL? What did the TBL experience mean to you?

TBL, team-based learning.

**Table 3 healthcare-10-00817-t003:** General characteristics of participants (N = 12).

Participant No.	Sex	Age	Self-Perceived Personality	Grade Point Average in the Last Semester *	Satisfaction with Nursing Major
1	F	20	Introvert	≥4.0	Satisfied
2	F	21	Introvert	<3.0	Satisfied
3	F	21	Extrovert	3.0–3.9	Satisfied
4	F	22	Ambivert	3.0–3.9	Satisfied
5	F	22	Ambivert	3.0–3.9	Moderate
6	F	22	Ambivert	3.0–3.9	Satisfied
7	F	20	Ambivert	3.0–3.9	Moderate
8	F	23	Extrovert	3.0–3.9	Satisfied
9	M	23	Ambivert	3.0–3.9	Satisfied
10	F	23	Introvert	<3.0	Moderate
11	F	21	Extrovert	3.0–3.9	Moderate
12	F	20	Extrovert	3.0–3.9	Satisfied

* The perfect grade point average is 4.5.

**Table 4 healthcare-10-00817-t004:** Themes and theme clusters.

Theme Clusters	Themes
Getting ready for learning	Constructive sense of burden that encourages learning
	Readiness in learning attitude and environment
Effective class promoting concentration and immersion	Identification of key learning points
	Repeating patterns for systematic and continuous learning
	Improvement in problem-solving abilities through reasoning of evidence
Proactive participation in a non-hierarchical learning environment	Differentiation with novelty and fun
	Flexible and comfortable class atmosphere
	Active and diverse interactions
Complementary collaboration	Accountability for the team
	Teamwork: A sense of being “one team”
Sense of burden	Burden from the role
	Learning burden

## Data Availability

Not applicable.

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
