# Peer review of "Team-Based Learning Experiences of Nursing Students in a Health Assessment Subject: A Qualitative Study"

_healthcare, 2022, doi:10.3390/healthcare10050817_

Round 1

Reviewer 1 Report

It is an interesting study focused on on the Team-based learning (TBL) experiences of nursing students in a health assessment subject. 
Relevant and necessary study, of interest to International Healthcare Review. Title is appropriate and informative. Objective should be the same in the abstract as in the main text, authors should review this  [p.2, l. 67-70]. The authors provide a good introduction to the topic of study, using recent and appropriate references as support. TBL should be better described for readers who have not used this teaching-learning methodology.  The need for this study is well founded, there is a lack of classroom experiences of students using this methodology. Objective well stated, however authors should avoid repetitive phrases [l. 67-71]. 
Materials and Methods: I have doubts that this design is phenomenological. Phenomenologies rely on philosophical theory (Husserl, Heidegger, Gadamer ...), use several interviews with the same participants and exploring their reflective process. This is descriptive or hermeneutics phenomenology? What were the criteria for inclusion of participants in the purposive sampling? [p. 2. l. 81-82] Adding a table explaining the TBL process applied to one of the systems would be of interest to readers. Where were the interviews conducted?, how did the authors ensure that students' responses were not affected by their position as students? Did the students have the possibility to learn with an alternative methodology or was TBL obligatory? Authors add a table of questions, this is a very positive contribution. Data analysis uses Colaizzi's method, this corresponds to descriptive phenomenology (see design section). How did they choose independent researchers to establish the rigour of the study?
Results: interesting, enjoyable to read. How the authors ensured that there was no competitive atmosphere between the groups?. In my opinion this can be good for learning because it involves the students [p. 6., l. 198-213]. Explain further how the methodology can improve practical skills, what skills do the students refer to? [pp. 6-7; l. 256-277]. What are the main differences between TBL and Problem Based Learning?
Discussion: Complete, very well worked, but very long discussion. Generally, no headings are used in this section of the scientific article. What do the authors propose to tackle the learning burden problems shown by the students? Limitations: how do the researchers tackle the fact that the experiences were contaminated by the teacher's evaluation of the subject? Conclusions respond to the objectives.

Reviewer 2 Report

Dear Authors,

Your research shows an interesting and important issue in nursing.

Despite the high value of your work I have a few doubts:

Line 49 - 51 However, there have been few studies on the essence of the TBL experiences from the students' perspectives, which limits the understanding of the TBL experience in the actual subjects of learning. Footnotes supporting this argument.

Line 82 - Twelve students out of a batch of 72 were finally selected. What did this selection consist of? For participants, what were the inclusion and exclusion criteria for eligibility?

Was the person conducting the interviews the teacher of these students?

Did the students receive any gratification for participating in the study? Were participants given incentives for participation?

Could participation in the study have had an impact on the credit of the subject?

I did not find any answers from the twelfth respondent. 
